# Hierarchical Reinforcement Learning for Power Network Topology Control

## Abstract

Learning in high-dimensional action spaces is a key challenge in applying reinforcement learning (RL) to real-world systems. In this paper, we study the possibility of controlling power networks using RL methods. Power networks are critical infrastructures that are complex to control. In particular, the combinatorial nature of the action space poses a challenge to both conventional optimizers and learned controllers. Hierarchical reinforcement learning (HRL) represents one approach to address this challenge. More precisely, a HRL framework for power network topology control is proposed. The HRL framework consists of three levels of action abstraction. At the highest level, there is the overall long-term task of power network operation, namely, keeping the power grid state within security constraints at all times, which is decomposed into two temporally extended actions: 'do nothing' versus 'propose a topology change'. At the intermediate level, the action space consists of all controllable substations. Finally, at the lowest level, the action space consists of all configurations of the chosen substation. By employing this HRL framework, several hierarchical power network agents are trained for the IEEE 14-bus network. Whereas at the highest level a purely rule-based policy is still chosen for all agents in this study, at the intermediate level the policy is trained using different state-of-the-art RL algorithms. At the lowest level, either an RL algorithm or a greedy algorithm is used. The performance of the different 3-level agents is compared with standard baseline (RL or greedy) approaches. A key finding is that the 3-level agent that employs RL both at the intermediate and the lowest level outperforms all other agents on the most difficult task. Our code is publicly available.

## 1 Introduction

Reinforcement learning (RL) has proven its worth in a series of artificial domains (Schrittwieser, 2020), and is beginning to show some successes in real-world scenarios such as robotics (Mahmood et al., 2018; Li et al., 2021), autonomous vehicles (Kendall et al., 2019), healthcare (Yu et al., 2021), and data centre automated cooling (Li et al., 2020). However, much of the research advances in RL are hard to leverage in real-world systems due to a series of assumptions that are rarely satisfied in practice (Dulac-Arnold et al., 2021). One of the key challenges is *learning and acting in high-dimensional state and action spaces*. These large state and action spaces can present serious issues for traditional RL algorithms.

Hierarchical Reinforcement Learning (HRL) is one method to address the "curse of dimensionality", enabling models that possibly scale RL to large state and action spaces (Chen et al., 2007; Al-Emran, 2015). In HRL, a RL task is decomposed into a hierarchy of *sub-tasks* which are generally easier to solve and their solutions might be reused to solve different problems (Barto & Mahadevan, 2003; Hengst, 2010; Pateria et al., 2021; Hutsebaut-Buysse et al., 2022). Efficiently using hierarchical decompositions has proven to make significant contributions towards solving various important open RL problems such as reward-function specification, exploration, sample efficiency, transfer learning, lifelong learning and interpretability (Hutsebaut-Buysse et al., 2022). This compositionality has even been identified as one of the key building blocks of artificial intelligence (AI) (Lake et al., 2017; Sutton & Barto, 2018; Eppe et al., 2022).

One crucial real-world task is power network control (PNC) which enables electricity to be a foundational element of modern life (Kelly et al., 2020). With the invention of electricity networks in the early 20th century, electricity can be considered one of the most important factors for the growth of economies and the functioning of societies around the world. It is a vital resource in daily living, industry, agriculture and public transportation. Blackouts of cities or countries, when they occur, are catastrophic events, which can cause chaos, and have a major impact on modern life. Hence, power networks represent critical infrastructures and power network operators, across the world, must ensure that a constant, reliable, secure, safe, cost-effective, and sustainable supply of electricity is maintained, while preventing blackouts at all times.

Controlling electricity - from generation sources to end-users' kettles - is an extremely complex task which is becoming even more challenging (ENTSO-E, 2021). To meet climate goals, power networks are relying more and more on less predictable renewable energy generation and decentralising their production. At the same time, the energy market is unified across Europe, and therefore calls for more interconnections and coordination between national grids. Moreover, networks are also ageing and infrastructure developments cannot keep up with the demand. These profound changes directly impact supervision systems in control rooms such that traditional tools used by electrical engineers to solve network issues are becoming increasingly inadequate (Panciatici et al., 2012; Marot et al., 2022b). In other words, PNC represents a crucial but complex sequential-decision making problem related to large state and action spaces. Hence, AI approaches are sought-after to help address the increasing uncertainty and the more numerous, complex, and coordinated decisions in order to derive optimal control actions (Marot et al., 2020b; Viebahn et al., 2022).

In this study, we showcase the potential of HRL for PNC. In the remainder of this section we briefly review the existing work on AI/RL applied to PNC (Sec. 1.1.1) as well as the relevant concepts of HRL (Sec. 1.1.2). In Section 2 we define more specifically the PNC problem and describe the power network environment considered in this study. In the central Section 3 we describe our HRL approach to PNC. Finally, after describing our experimental setup in Section 4 we analyze the results of several HRL agents acting in the power network environment for different experimental settings in Section 5. We end with Section 6 providing conclusions and directions for future work.

## 1.1 Related work

In this subsection we briefly review the existing work on AI/RL applied to PNC (Sec. 1.1.1), and subsequently we summarize the relevant concepts of HRL (Sec. 1.1.2).

### 1.1.1 Reinforcement learning applied to power network control

Research on sequential decision making applied to real-time power network control is still in its infancy. The opportunities for scientists to work collaboratively at scale on the problem were limited by a lack of commonly usable environments, baselines, data, networks, and simulators. However, the ongoing energy transition forces industry and academia to invest significant resources in this topic. Recently, RTE TSO developed the open-source GridAlive ecosystem (RTE France, 2020b) to facilitate the development and evaluation of controllers (or agents) that act on power grids. With the Grid2Op framework (RTE France, 2020a) at its core, any type of control algorithm in interaction with simulators of one's choice can be used such that gaps between research communities can be overcome. In particular, it enables casting the power grid control problem into the framework of a Markov decision process (MDP) (Sutton & Barto, 2018).

Based on the GridAlive ecosystem, RTE TSO and collaborators launched a series of competitions, the so-called *Learning to Run a Power Network* (L2RPN) challenge (RTE France; Marot et al., 2020a; 2021; 2022a; Serre et al., 2022). In each competition the participants need to develop controllers that operate a power network to maintain a supply of electricity to consumers on the network over a given time horizon by avoiding a blackout. The controllers are exposed to realistic (stochastic) production and consumption scenarios, and the remedial actions are subject to real-world network constraints. The aim of L2RPN is to foster faster progress in the field by creating the first large open-benchmark for solutions to the real-world problem of complex continuous-time network operations, building on previous advances in AI such as the ImageNet benchmark for computer vision (Deng et al., 2009).

Initial L2RPN competitions tested the feasibility of developing realistic power network environments (Marot et al., 2020a) and the applicability of RL agents (Lan et al., 2020; Yoon et al., 2020; Subramanian et al., 2021; Matavalam et al., 2022). Subsequently, the 2020 L2RPN competition at NeurIPS (Marot et al., 2021; Zhou et al., 2021), the 2021 *L2RPN with trust* competition (Marot et al., 2022a), and the 2022 L2RPN competition for carbon neutrality (Serre et al., 2022; Dorfer et al., 2022) had increased complexity. These competitions came with more realistically sized network environments, implying very large discrete action spaces due to the combinatorial topological flexibilities of the power network. Moreover, three real-world network operation challenges are addressed, namely, robustness, adaptability, and trustworthiness. In the robustness track, controllers had to operate the network maintaining supply to consumers and avoiding overloads while targeted unforeseeable line disconnections create challenging situations by disconnecting one of the most loaded lines at random times. In this study we also focus on the robustness challenge (see Section 4.1.2).

Throughout the competitions, ML approaches showcased continuous robust and adaptable behaviours over long time horizons. This behaviour was not previously exhibited by the expert systems (Marot et al., 2018), or by optimization methods that are limited by computation time (Ruiz et al., 2017; Little et al., 2021). Participation and activity were steady with entries from all over the world, and corresponding research is emerging (Lan

et al., 2020; Yoon et al., 2020; Subramanian et al., 2021; Matavalam et al., 2022; Zhou et al., 2021; Dorfer et al., 2022). The winning solutions employ a combination of expert rules, brute force simulation for action validation, and on top of that different RL approaches to increase planning abilities and get a final boost in operational performance. In particular, RL agents based on Proximal Policy Optimization (PPO) (Schulman et al., 2017) combined with previously reduced action spaces can be considered as state-of-the-art since such an approach has been employed in high-ranking competition submissions (see 2nd place in Marot et al. (2021)) and subsequent research (Lehna et al., 2023; Chauhan et al., 2023).

Finally, we again emphasise that research on sequential decision making applied to real-time power network control is still at its very beginning. Until now even the best agents still fail over 30% of the L2RPN test scenarios and sending alarms based on confidence levels is equally successful. Moreover, currently available approaches are often partly tweaked for specific competition setups. For example, successful approaches often employ specific hard-coded sets of topologies which have been identified by the participants in an exploratory phase preceding the actual model training phase. Consequently, much more systematic benchmarking of RL approaches applied to power network control is necessary, and clean approaches that employ generic techniques are needed (see e.g. Subramanian et al. (2021)).

### 1.1.2 Hierarchical reinforcement learning

In HRL, a RL task is decomposed into a hierarchy of *sub-tasks* such that higher-level parent-tasks invoke lower-level child tasks as if they were primitive actions (Barto & Mahadevan, 2003; Hengst, 2010; Pateria et al., 2021; Hutsebaut-Buysse et al., 2022). A decomposition may have multiple levels of hierarchy, and the hierarchy of policies collectively (i.e. the so-called *hierarchical policy*) determines the behavior of the agent. Some or all of the sub-tasks can themselves be RL tasks. That is, HRL also includes hybrid approaches such that, for example, planning at the top level can be combined with RL at the more stochastic lower levels (Reid & Ryan, 2000). In most applications the structure of the hierarchy is provided as background knowledge by the designer. In such approaches the programmer is expected to manually decompose the overall problem into a hierarchy of sub-tasks.

In order to specify sub-tasks different types of *abstractions* can be used, allowing agents to build up higher-level actions from lower ones. Often *temporal abstractions* are used that lead to sub-task policies that persist for multiple time-steps and are hence referred to as *temporally extended actions*. In this case the task decomposition effectively reduces the original task's long horizon into a shorter horizon in terms of the sequences of sub-tasks. This is because each sub-task is a higher-level action that persists for a longer timescale compared to a lower-level action. Such an approach changes the MDP setting to a *Semi-Markov decision process* (SMDP) (Sutton et al., 1999; Baykal-Gursoy, 2010). Moreover, *state abstractions* are often used since typically only specific aspects of the state-space are relevant for specific sub-tasks. Ignoring features irrelevant to the sub-behavior allows for more efficient learning.

One popular approach in HRL is the *options framework* (Sutton et al., 1999; Hutsebaut-Buysse et al., 2022). Options are temporally extended actions that can be defined as a tuple $\omega = (\mathcal{I}_\omega, \pi_\omega, \mathcal{T}_\omega)$ consisting of an initiation condition[1] $\mathcal{I}_\omega$, a termination condition $\mathcal{T}_\omega$, and the intra-option policy $\pi_\omega$. Using a well-defined set of options will require the agent to make fewer decisions when solving problems, and consequently, can speed up learning (Silver & Ciosek, 2012; Mann & Mannor, 2014). Different methods of training options are possible. Options can be trained to maximize attainment of some local goal, achieving recursive rather than hierarchical optimality (a weaker solution concept; Dietterich (2000)). However, end-to-end training optimizes performance in the overall task Bacon et al. (2017). Such learning strategies typically do not explicitly learn an initiation set (instead letting the initiation set be the complete state space). Those methods rely on the policy-over-options to select an appropriate option in each state (Harb et al., 2018).

## 2 Power System Environment

In this section we first describe the power network control problem in general terms (Sec. 2.1). Subsequently, we describe the concrete example used in this study including the power grid model (Sec. 2.2), the objective including the relevant constraints (Sec. 2.3), and the related state and action spaces (Sec. 2.4). Finally, we define the reward function employed in this study (Sec. 2.5).

### 2.1 The power network control problem

Control centres are core places of the power system, providing to groups of human operators the necessary working environment to remotely monitor and operate the power system in real time (Marot et al., 2022b).

---

[1]Often the initiation condition is simply that the current state belongs to a pre-defined initiation set.

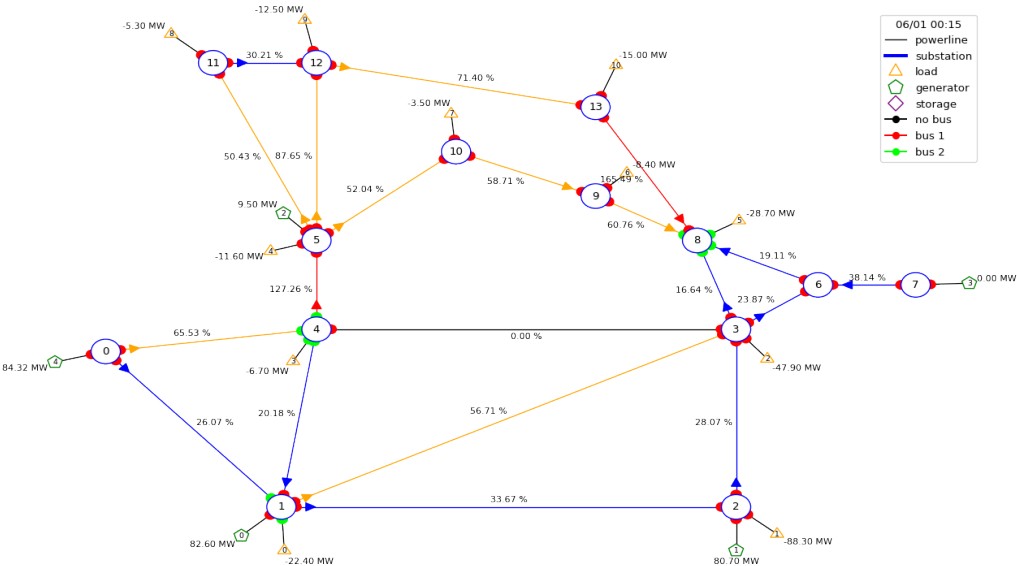

Figure 1: Standard visualization of the IEEE 14-bus network in *Grid2Op* (RTE France, 2020a). The network consists of 14 substations (blue circles), 20 lines, 11 loads (yellow triangles), 5 generators (green pentagons). The active power of each load and generator is given (in MW) as well as the resulting loading of the lines (in %). Each substation actually consists of two busbars, and for a substation each connected element (i.e. a line, load or generator) is connected to one of the two busbars which is indicated by either a red (busbar 1) or a green (busbar 2) half-circle. In this example in substations 1, 4, and 8 elements are connected to busbar 2 (i.e. green half-circles appear). Moreover, the line connecting substations 3 and 4 is out of service (due to overloading).

The operators interact with the power system, on the one hand, by observing the continuously changing power system state, and, on the other hand, by manually performing a broad range of control actions on the grid. The actions can be discrete like line switching and substation reconfiguration, or continuous like adjusting voltage setpoints or power dispatches of generators, and many more (Kelly et al., 2020; Viebahn et al., 2022).

More specifically, secure operation of power networks is required both in normal operating states as well as in contingency states (i.e. after the loss of any single element on the network). That is, the following requirements must be met: (*i*) In the normal operating state, the power flows on equipment, voltage and frequency are within pre-defined limits in real-time. (*ii*) In the contingency state the power flows on equipment, voltage and frequency are within pre-defined limits. Loss of elements can be anticipated (scheduled outages of equipment) or unanticipated (faults for lightning, wind, spontaneous equipment failure). Cascading failures must be avoided at all times to prevent blackouts (corresponding to the game over state in the RL challenge) (Kelly et al., 2020).

Importantly, each of the control actions performed by power system operators usually not only affects the current state of the power system but also the future state and availability of future control actions, that is, short-term actions can have long-term consequences. As a result, the decision problem of power system operators is typically a **sequential decision-making problem** in which the current decision can affect all future decisions. Moreover, due to possible nondeterministic changes of the power system state (e.g., due to unplanned outages or the intermittent behaviour of renewable energy sources) and different sources of error (e.g., measurement errors, state estimation errors, flawed judgement) the operators need to **handle uncertainty** in their decisions. Finally, operational decisions must often be made quickly, under **hard time constraints**, and as mentioned in Sec. 1.1.1, the **action space is very large** due to the combinatorial flexibilities of the power network (Viebahn et al., 2022).

The energy transition increases both the relevance of this challenge as well as the complexity of time horizons, uncertainty, and time constraints in control centres. The traditional tools used by electrical engineers to solve network issues are becoming increasingly inadequate. Variations in demand and production profiles, with increasing renewable energy integration, as well as the high voltage network technology, constitute a real challenge for human operators when optimizing electricity transportation while avoiding blackouts. AI approaches (in particular RL) are well-suited to support the evolution of control centres since AI in large parts inherently deals with sequential decision making under uncertainty. In other words, AI is a logical candidate for designing advanced decision support tools for power system operators with the aim to improve the efficiency and safeguard the reliability, security, and resilience of power systems (Viebahn et al., 2022)

## 2.2 Power grid model used in this study

The power grid model considered in this study is a slightly adapted version of the IEEE 14-bus network, as it was created for the L2RPN challenge 2019 (RTE France). Figure 1 sketches the main elements of the grid: 14 substations (blue circles), 20 lines connecting the substations, 11 loads (yellow triangles) and 5 generators (green pentagons). Generation includes a wind power plant and a solar in-feed next to a nuclear generator and two thermal generators to represent the current energy mix. We use the thermal current limits of the lines as described in Subramanian et al. (2021) to make the difference between the transmission grid (i.e. higher voltage, substations 0-4) and the distribution grid (i.e. lower voltage, substations 5-13) more pronounced. The transformers to step down the voltage from the transmission side to the distribution side are modelled as lines (connecting substations 4 and 5, and substations 3, 6, 8). Finally, we note that each substation actually consists of two busbars[2]. For a substation each connected element (i.e. a line, load or generator) is connected to one of the two busbars which is indicated in Fig. 1 by either a red (busbar 1) or a green (busbar 2) half-circle. *Importantly, the control actions considered in this study are given by substation re-configurations in which the busbar to which an element in a substation is connected can be changed* (see also Sec. 2.4 below). Obviously, this changes the topology of the network, that is, the way the different elements of the network are connected with each other. Note that in the *default configuration* for each substation all elements are connected to busbar 1.

The power grid model is available within the Python package *Grid2Op* (RTE France, 2020a) that provides an easy to use framework for the development, training, and evaluation of 'agents' or 'controllers' that act on a power grid. It uses *Pandapower* (Thurner et al., 2018) as a backend[3] for power flow computations and the package is compatible with Open-AI gym (OpenAI).

The module is accompanied by datasets that represent realistic time-series of operating conditions. The dataset for the IEEE 14-bus model encompasses 1,000 scenarios, each containing data for 28 continuous days at 5-minute intervals. Each scenario incorporates predetermined load variations and generation schedules that, in combination with the grid topology, determine the power flow at every time step. These schedules reflect the load and generation distribution of the French power grid (Marot et al., 2020a). A scenario, combined with the agent's actions, constitutes an episode. An episode persists for 8,064 steps or until a game over condition is encountered (see Sec. 2.3).

## 2.3 Objective and constraints

The overall objective is to create an agent that is able to operate the power grid successfully (i.e. keeping the power grid state within security constraints) for as many scenarios as possible, using only substation reconfiguration actions (see Sec. 2.4 for details). In doing so, the agent must respect a number of operational constraints (RTE France, 2020a). These consist of *hard constraints* which trigger an immediate "game over" condition if violated, namely:

(a) system demand must be fully served;

(b) no generator may be disconnected;

(c) no electrical islands are formed as a result of topology control;

(d) a solution of the power flow equations must exist at all times.

In contrast, *soft constraints* have less severe consequences: Transmission lines with a current exceeding 150% of their rated capacity are tripped immediately, and can be recovered after 50 minutes (10 time steps). When lines are overloaded by a smaller amount, the agent has 10 minutes (2 time steps) to mitigate this. If lines remain overloaded after this time, they are disconnected and are not reconnected until the end of the episode. In addition, substations are subject to a practical 'complexity' constraint that only one substation can be modified per timestep and a 'cooldown time' (15 minutes) needs to be respected before a switched node can be reused for action. Both soft and hard constraints make the problem more practical and close to real-world grid operation.

---

[2]A busbar is a metallic strip or bar that is typically housed inside a substation. A busbar is a type of electrical junction (or node or connection point) in which all the incoming and outgoing electrical current meets. A popular substation setup includes two busbars. This setup provides multiple options to connect the elements connected to a substation (e.g. lines, generators, loads) by connecting each element to only one of the two busbars. Dependent on the specific substation configuration the electrical current can be routed in different ways.

[3]The backend corresponds to a power flow computation-based action simulator. Given an action and an observation, it simulates the state of the network and the reward in the next time step. Though accurate, the computational cost incurred is prohibitive if one were to simulate the whole action space on large networks.

| Name | Size | Description |
|------|------|-------------|
| Active power | $N_{gen} + N_{load} + 2N_{line}$ | Active power magnitude. |
| $\rho$ | $N_{line}$ | The loading of each power line, which is defined as ratio between current flow and thermal limit. |
| Topology Configuration | $N_{gen} + N_{load} + 2N_{line}$ | For each element (load, generator, ends of a line), it gives on which bus these elements is connected in its substation. |
| Time step overflow | $N_{line}$ | The number of time steps each line is overflowed. |

Table 1: State features used to model the state.

## 2.4 State space and primitive action space

In general, the state of the power network at each time step is given by a snapshot of measurements (related to physical properties of the network elements) and the topology of the network (i.e. the switch states of the network elements). In this study, we use the state features presented in Table 1 (which are similar to what is used in e.g. Yoon et al. (2020)). These include the loading of the lines, $\rho$, and the amount of active power produced by the generators, consumed by the loads, and transmitted through the power lines. Moreover, the topology of the network is encoded via the busbar assignment of each element. Finally, the number of timesteps a line is overloaded is also included.

Regarding the action space, in this work we only consider one type of primitive action, namely, *substation reconfiguration*, that is, changing the busbar to which an element is connected in a given substation (see Fig. 1 for an example). Effectively, the agent's primitive action space consists of all possible substation configurations (i.e. all possible ways to put the connected elements either to busbar 1 or to busbar 2). Hence, if we denote the number of substations by $N_{sub}$, the number of elements in the $i$-th substation by $Sub(i)$, then in general an agent at time $t$ can choose an action $a_t \in \mathcal{A}$ with $|\mathcal{A}| = \sum_{i=1}^{N_{sub}} 2^{Sub(i)}$. However, we further restrict the primitive action space to adhere to the static constraints outlined in Subramanian et al. (2021) (see also Sec. 2.3) which reduces the action space from 248 to 112 substation configurations. Finally, we (*i*) remove the configurations for substations where only one configuration is possible (i.e. no substation reconfiguration is possible), and (*ii*) we add one explicit "do-nothing" action. Consequently, we are left with 106 substation configurations (i.e. primitive actions) across 7 out of 14 substations. Note that the combination of substation configurations results in over 23 million possible grid topologies (Subramanian et al., 2021).

## 2.5 Reward

The topic of suitable reward functions in the context of power network control is an important question that requires further research. However, this topic is not the focus of this study and, hence, we employ the most commonly used reward function as introduced by Marot et al. (2020a). This reward function is designed to measure the overall strain on the grid at each timestep, thus promoting better performance compared to simple step-constant rewards. Let $N_{lines}$ denote the number of lines, $F_i$ the electricity flow in amps in powerline $i$, and $L_i$ the thermal limit of line $i$. The margin of powerline $i$ can be computed as:

$$M_i = \frac{L_i - F_i}{L_i} \text{ if } F_i \leq L_i \text{ else } 0 .$$

The reward is then computed via the following formula:

$$\frac{1}{N_{line}} \sum_{i=1}^{N_{line}} 1 - (1 - M_i)^2 .$$

This reward function encourages the agent to keep the load on the powerlines below the thermal limit and maintain a uniform load distribution among lines. The scaling factor ensures that the reward remains bounded between 0 and 1 (this is essential for some RL algorithms such as e.g. SAC, see B.2).

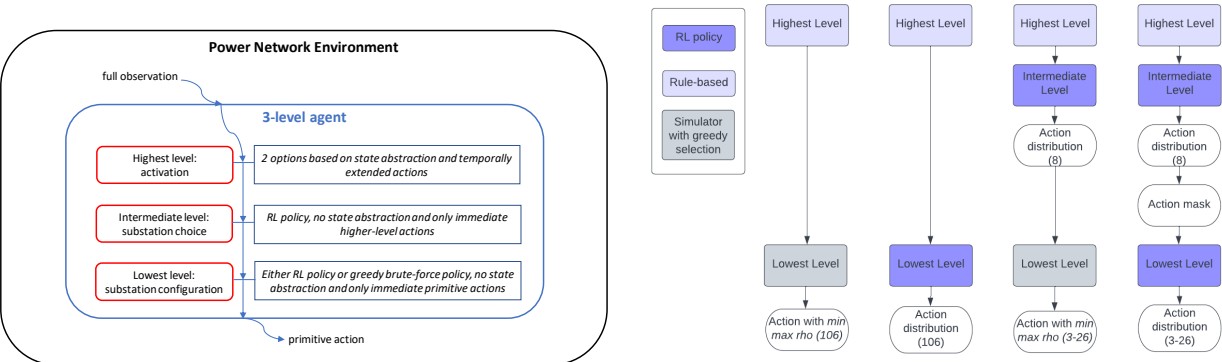

Figure 2: **Left:** Hierarchical structure of the 3-level power network agents considered in this study. A full observation is received from the power network environment and processed at different levels. The red boxes indicate the different hierarchical levels, and the connected blue boxes indicate the HRL concepts employed at each level. Finally, a primitive action is executed in the power network environment. **Right:** Schematic overview of different kinds of agents. The number of simulated actions and dimension of the probability distribution is indicated in parenthesis.

## 3    Power Network Agents

In this section we describe our HRL approach to PNC. First we define our 3-level HRL framework (Sec. 3.1). Subsequently, we describe the different agents analysed in this study which differ in the number of employed hierarchy levels and the type policy used in each level (Sec. 3.2).

### 3.1    Hierarchy of sub-tasks

All power network agents generally perform the same main long-horizon task, namely, keeping the power grid state within security constraints at all times. In hierarchical modeling, the main task is decomposed into different sub-tasks. In this study, the different sub-task are handcrafted and the aim is to learn a hierarchical policy. This is a non-trivial challenge because any approach to learn a hierarchical policy must tackle the following key issues: carefully designing the algorithm to learn the policies at various levels of the hierarchy (including reward propagation, value function decomposition, state/action space design, etc.), dealing with non-stationarity due to simultaneously changing policies, ensuring the optimality of the hierarchical policy as a whole, interpretability of the hierarchical policy, among other issues (Pateria et al., 2021).

In this study, we investigate the decomposition of the main long-horizon task into three levels, as schematically shown in Fig. 2 (left). The highest level concerns the activation of the agent and is based on two options (Sec. 3.1.1). At the intermediate level the agent performs a substation choice which is given via a RL policy (Sec. 3.1.2). And at the lowest level the agent identifies a suitable substation configuration (Sec. 3.1.3).

#### 3.1.1    Highest level: Activity threshold reformulated in the options framework

All control agents can be faced with the high-level question: Do I need to act or should I better do nothing? The solutions developed for the L2RPN challenge generally solve this problem by employing a so-called activity threshold (Subramanian et al., 2021), that is, an agent only executes actions if the current highest line loading in the power grid exceeds a certain fixed threshold $\rho_{thres}$, otherwise the grid topology remains unchanged (i.e., a do-nothing action is chosen). Applying an activity threshold significantly improves the learning behaviour since learning of appropriate behaviour in low-loading situations can be omitted. The threshold $\rho_{thres}$ is usually chosen to be between 80% and 100% (Yoon et al., 2020; Marot et al., 2020a).

This approach can be described as a binary decomposition of the main task into two sub-tasks, namely, either 'don't act as long as it's save' or 'reduce the highest line loading'. Both sub-tasks can last for several timesteps and, hence, generally represent temporally extended actions of varying duration. Moreover, a state abstraction is employed since not the entire state of the power network but only the highest line loading, $max(\rho)$, is used as input. Conceptualized via the options framework, we have two options $\omega_1 = (\mathcal{I}_{\omega_1} : \max(\rho) < \rho_{thres}, \pi_{\omega_1} : \,'\text{do nothing}', \mathcal{T}_{\omega_1} : \max(\rho) \geq \rho_{thres})$ and $\omega_2 = (\mathcal{I}_{\omega_2} : \max(\rho) \geq \rho_{thres}, \pi_{\omega_2} : \,'\text{reduce } \max(\rho)', \mathcal{T}_{\omega_2} : \max(\rho) < \rho_{thres})$. As mentioned in Sec. 1.1.2, such an approach changes the MDP setting to the more general SMDP setting because the state transitions can occur in irregular time intervals (in other words, one has to deal with

temporally extended actions of varying duration). In our case specifically, the do-nothing action is executed by default if $\max(\rho) < \rho_{thres}$. As usual in SMDP settings, an aggregate reward is computed from all do-nothing actions between policy-driven steps. Modeling the interaction as an SMDP in this fashion avoids evaluating the policy and calculating updates for many low-impact states, and thus improves computational efficiency.

In this study, all agents employ the same binary options framework at the highest hierarchy level with $\rho_{thres} = 95\%$. The exact intra-option policy $\pi_{\omega_2}$ is then created at lower hierarchy levels. We note that in future research this approach can be extended, for example, by designing more options or by learning a policy-over-options (see Sec. 1.1.2).

### 3.1.2   Intermediate level: substation choice

At the intermediate hierarchy level, a policy is trained via RL with the intermediate-level action space given by the set of controllable substations. As outlined in Sec. 2.4, there are 7 controllable substations in the power system environment which is an order of magnitude smaller than 106 relevant substation configurations (i.e. primitive actions). Moreover, the full observation as described in Sec. 2.4 (i.e. no state abstraction) and the reward as described in Sec. 2.5 are used. We note that a substation choice is an immediate (i.e. not temporally extended) intermediate-level action. In future work, the intermediate-level action space can be enlarged by temporally extended actions corresponding to sets (and subsequently sequences) of substations.

### 3.1.3   Lowest level: substation configuration

The lowest hierarchy level consists of the primitive actions, that is, substation configurations. As outlined in Sec. 2.4, the plain primitive action space consists of all possible substation configurations (106 in case of the IEEE 14-bus network). For agents that employ the intermediate hierarchy level (Sec. 3.1.2) the set of substation configurations is reduced to the possible configurations of the substation chosen at the intermediate level. Hence, then the size of the primitive actions space varies dependent on the chosen substation (3-26 instead of 106 for the IEEE 14-bus network). Again, the full observation as described in Sec. 2.4 (i.e. no state abstraction) and the reward as described in Sec. 2.5 are used.

## 3.2   Hierarchical power network agents

Figure 2 (right) schematically depicts the different agent architectures employed in this study. All agents share the same task decomposition (and even the same rule-based policy) at the highest hierarchy level (Sec. 3.1.1). The agents differ with respect to the subsequent task decompositions which can be either 3-level approaches (Sec. 3.2.1) or simpler 2-level approaches (Sec. 3.2.2). Moreover, for each approach we consider different types of policies. More specifically, for policies trained via RL we compare the performance of two state-of-the-art RL algorithms, namely, Proximal Policy Optimization (PPO) (Schulman et al., 2017) and Soft Actor Critic (SAC) (Haarnoja et al., 2018a). This choice is motivated by the availability of an implementation in the *RLlib* framework (Liang et al., 2018) and by the fact that each algorithm is suited to deal with both discrete and continuous large action spaces. And as already noted in Section 1.1.1, RL agents based on PPO combined with previously reduced action spaces can be considered as state-of-the-art since such an approach has been employed in high-ranking competition submissions (see 2nd place in Marot et al. (2021)) and subsequent research (Lehna et al., 2023; Chauhan et al., 2023).

### 3.2.1   3-level approaches

In this study, we consider three different 3-level agents (see also Fig. 2 (right)): (*i*) In *PPO Substation* the intra-option policy $\pi_{\omega_2}$ at the intermediate level is trained using the state-of-the-art RL algorithm PPO (Schulman et al., 2017), whereas at the lowest level a greedy brute-force approach is applied. More precisely, at each timestep for a given substation (which is chosen at the intermediate level) all possible primitive actions (i.e. configurations) related to that substation are simulated for the next timestep and the action that reduces $\rho$ the most is chosen. In short, in this approach an RL policy at the intermediate level is combined with greedy brute-force optimization at the lowest level. Similarly, (*ii*) in *SAC Substation* the intra-option policy $\pi_{\omega_2}$ is trained using the RL algorithm SAC (Haarnoja et al., 2018a), whereas at the lowest level the same brute-force approach as in *PPO Substation* is used. Finally, (*iii*) in *PPO Hierarchical* the intra-option policy $\pi_{\omega_2}$ is entirely based on RL with two separate policies on the intermediate level and the lowest level that are both trained via PPO. We choose to only apply PPO in the case with RL at two levels because the results using SAC where less promising (see Sec. 5).

Regarding *PPO Hierarchical* a few more modeling details are relevant: The *PPO Hierarchical* model is trained with two separate policies for which the actor parameters on different levels are not shared. In contrast, the parameters of the value function are shared which we found to work better than sharing no parameters.

Moreover, given a full observation (see Sec. 2.4), the intermediate-level policy chooses the substation. The one-hot-encoded substation choice is concatenated with the observation and is used as input to the lowest-level policy. The substation choice is also used to create an action mask. The action mask adds a large, negative number to the policy logits of all actions that do not correspond to the actions at a given substation. Finally, given the action sampled from the lower-level policy a step in the environment is taken, yielding a new observation and reward. The same reward signal is given to both policies.

### 3.2.2 2-level approaches

In order to benchmark the 3-level hierarchical agents we compare with simpler approaches in which the primitive action space is not reduced via a hierarchical approach. In these 2-level approaches only the highest (Sec. 3.1.1) and the lowest (Sec. 3.1.3) hierarchy levels are employed. Consequently, the primitive action space of the lowest level is not reduced and contains 106 substation configurations (see Sec. 2.4).

More precisely, we consider three variations of this 2-level approach (see also Fig. 2 (right)): (*i*) In *PPO Native* the intra-option policy $\pi_{\omega_2}$ is trained using the state-of-the-art RL algorithm PPO. (*ii*) In *SAC Native* the intra-option policy $\pi_{\omega_2}$ is trained using the RL algorithm SAC. Finally, (*iii*) the *Greedy Expert* represents a popular baseline brute-force approach in which at each timestep all possible primitive actions are simulated for the next timestep and the action that reduces $\rho$ the most is chosen. In other words, *Greedy Expert* represents a purely rule-based approach that does not take any time horizon into account (it is 'greedy').

## 4 Experimental setup

In this section we make our experimental setup more precise. We begin by describing the two experimental regimes that we investigate (Sec. 4.1), followed by addressing the evaluation process (Sec. 4.2) and relevant hyperparameters (Sec. 4.3).

### 4.1 Experimental regimes

In this study, we investigate two different experimental regimes which correspond to two different difficulty levels (or equivalently, different levels of realism) of the power system environment. For that recall from Sec. 2.1 that in the real world secure operation of power networks is required both in normal operating states as well as in contingency states (i.e. after the loss of any single element on the network).

#### 4.1.1 Less realistic regime: Power system environment without contingencies

In order to establish a baseline, we first train, validate, and test our models in an environment devoid of line outages (the only line disconnections occur due to overloading for consecutive time steps). That is, secure operation in contingency states is neglected in this regime. This setup allows us to observe the agent's performance in an idealized setting. In this case power network operation is less challenging such that only agent architectures that perform rather well in this regime are suitable for further analysis and development in more difficult regimes.

#### 4.1.2 More realistic regime: Power system environment with contingencies

In order to deal with more realistic circumstances, we introduce *random* line outages into the power system environment to simulate real-world challenges that agents operating power grids must overcome. These outages simulate real-world contingencies that stem from external factors such as weather conditions or equipment attrition. More precisely, we model the outages by implementing the following set of rules that dictate the disconnection and reconnection of lines, as well as the frequency and feasibility of these events:

- Exactly one line is disconnected

- Disconnected lines are sampled uniformly

- After exactly 4 hours the line is re-connected

- There are 2 outages per day

- To make the grid operation feasible, only specific power lines can be disconnected. These are the power lines connecting the following substation pairs: 3-4, 3-6, 3-8, 6-8, and 11-12 (see Fig. 1).

This modification allows us to revisit the training and performance of our models under more realistic circumstances.

### 4.2 Evaluation

To arrive at a realistic estimate of the agents' performance we select 70% of the scenarios as a training set, 10% as a validation set, and 20% as a testing set. As a reminder, a scenario is a time series of prescribed load and generation variations. Each scenario contains 8064 time steps (see Sec. 2.2). We ensure that the difficulty level of scenarios is balanced over the different sets in the following way: We approximate the difficulty level of a scenario by the number of actions taken by the Greedy Expert agent (defined in Sec. 3.2.2) in that scenario. Subsequently, we create 10 buckets with each containing 100 scenarios of a similar difficulty level. Finally, we perform a 70-10-20 split of each bucket followed by a concatenation of the 70-10-20 splits.

In the upcoming results section (Sec. 5), our primary metric of interest is the number of steps survived throughout the tested scenarios, reflecting the agent's ability to operate the grid without catastrophic failure. Moreover, we consider the reward normalized by the number of survived time steps, which reflects how efficiently the agent can operate the grid. In appendix A, we present a more detailed analysis and investigate also metrics such as topological depth (i.e. the number of substations that are not in default configuration), and agents' activity levels to gain a more comprehensive understanding of the agent's behaviour.

### 4.3 Hyperparameters

All models use $\rho_{thres} = 95\%$ below which the do-nothing action is executed. For all models, we use $N = 3$ MLP layers of 256 units as the policy and value function approximators. The parameters between the value function and the policy are not shared. For PPO we use a batch size of 1024 with a mini-batch size of 256. For SAC we use the batch size of 512. The learning rate $\eta = 10^{-4}$ for both actor and the critic is used for models using SAC and $\eta = 5 \times 10^{-5}$ for models using PPO. We found that higher learning rates result in more frequent divergence of the models, though some models still converged. Further details about the parameters are available in the appendix, Section B. The *RLlib* framework (Liang et al., 2018) was used for efficient, parallelized training. Our code in PyTorch (Paszke et al., 2017) is publicly available at https://anonymous.4open.science/r/topologyControlHrl/.

## 5 Results

In this section, we report on the training stability and performance of different hierarchical power network agents (as defined in Sec. 3.2) in both experimental regimes (as defined in 4.1). The agents with a suffix *Native* are described in Section 3.2.2, with a suffix *Substation* or *Hierarchical* in Section 3.2.1. The schematic overview of different agent types is portrayed in Figure 2 (right).

### 5.1 Training stability and performance

Figure 3 shows the mean episode return (solid line) and the corresponding standard error (shaded area) averaged over different model seeds (12 in total). In the experimental regime without contingencies (upper-left panel) all PPO models reach a rather high performance level. More specifically, the PPO agents exhibit smaller variance between seeds, have faster convergence properties, and obtain higher expected rewards compared to the models using SAC. Despite extensive hyperparameter tuning (see appendix B), similar behavior could not be achieved for the models using SAC. The SAC Substation model was the hardest to train, achieving subpar results for the majority of the runs. Only 3 of the 10 models achieve validation performance above 7700 mean episode length, with the majority of the models saturating around 7000. The SAC Native model improved stably but converged at only slightly higher levels of the mean episode return and shows the largest variance.

In the experimental regime with contingencies (Figure 3 upper-right panel) the training behaviour is significantly different. Firstly, the models converge only after 5 to 10 times more episodes at around only half of the mean episode return compared to the regime without contingencies. The SAC Substation model achieves a significantly lower mean return than the other models. Similar to the training without contingencies, the PPO Substation stabilizes the fastest and is characterized by the smallest variance in performance between seeds. In contrast, PPO Hierarchical is characterized by the highest variance. The corresponding standard deviation has a magnitude of 1500 such that some seeds lead to the highest and other seeds lead to the lowest mean return of all agents. This is a result of a dichotomous convergence behavior of PPO Hierarchical shown in the lower-right panel of Figure 3. We see two groups: the "performant" group that after around 300 environment interactions steeply improves but has high variance, and the "non-performant" group that stays at subpar return levels but is more stable. In other words, a bifurcation emerges in hierarchical policy space when training on different random seeds. For agents employing RL at only one level the hierarchical policies (due to different seeds) are unimodally distributed around the mean return. Only when RL is employed at two levels a bimodal distribution of hierarchical policies emerges such that the set of hierarchical policies related to one mode outperforms all other approaches. The lower-left panel of Figure 3 shows the training behavior compared to other agents when

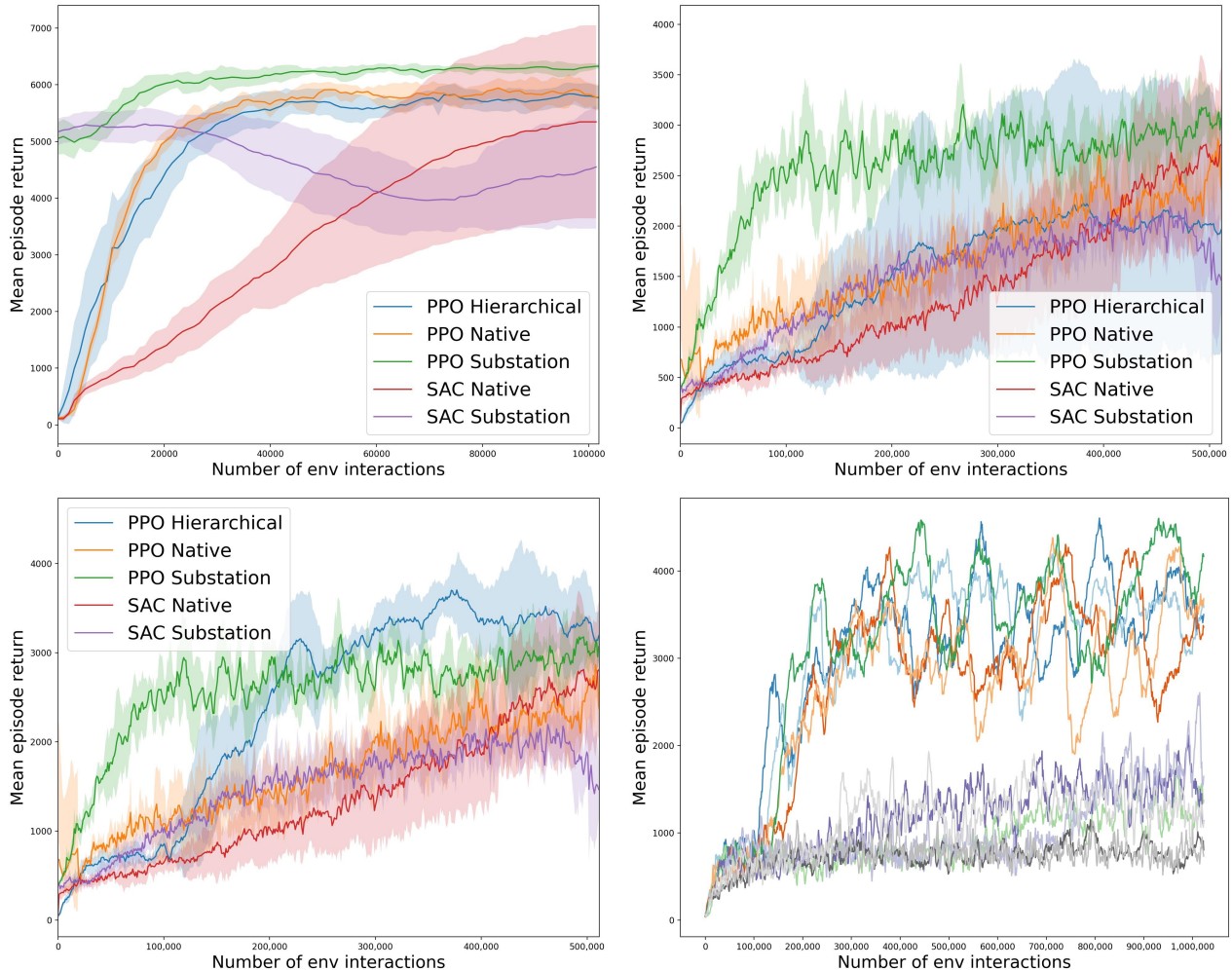

Figure 3: Training curves of different hierarchical RL agents. The upper-left panel is related to the experimental regime without contingencies whereas the other three panels are related to the regime with contingencies. The mean episode return on the training set averaged over 12 different training seeds per agent is shown. The shaded area denotes the standard error. Upper-right panel: For PPO Hierarchical the models of all training seeds are included. Lower-left panel: For PPO Hierarchical only the models of the performant cluster are included. Lower-right panel: Individual training curves related to the different training seeds of the PPO Hierarchical agent depicting dichotomous convergence behavior leading to two clusters of models.

taking only the performant group into account which exhibits the highest mean episode return of all agent types. Note that in appendix C extended training curves for the PPO models are shown.

During training of PPO Hierarchical, the two PPO policies depend on each other which poses optimization challenges. If the higher-level policy chooses a substation where no action can prevent a catastrophic failure both policies are penalized the same way as they receive the same reward signal. Moreover, the higher-level policy must decide on the lower-level policy which is changing in course of optimization, making this a non-stationary problem for the higher-level policy. This causes instabilities and big variance in the optimization process. Making hierarchical training more stable and sample efficient is an active research area. Approaches include crafting rewards based on prior domain knowledge (Kulkarni et al., 2016) (Heess et al., 2016) or employing goal relabeling techniques (Nachum et al., 2018). However, this is out of scope for this work.

## 5.2 Generalization to unseen scenarios

For each agent type the model with the highest mean episode length on the validation set is chosen for further analysis[4]. Figure 4 shows the performance (i.e. the mean episode length) as well as the agents' activity level for of the best model per agent type averaged over the 200 scenarios of the test set.

---

[4]Note that similar performance differences between agent types are seen in the model mean, as indicated in Fig. 3, as well as when evaluation is done on all scenarios, as shown in tables 2 and 3 in appendix A.

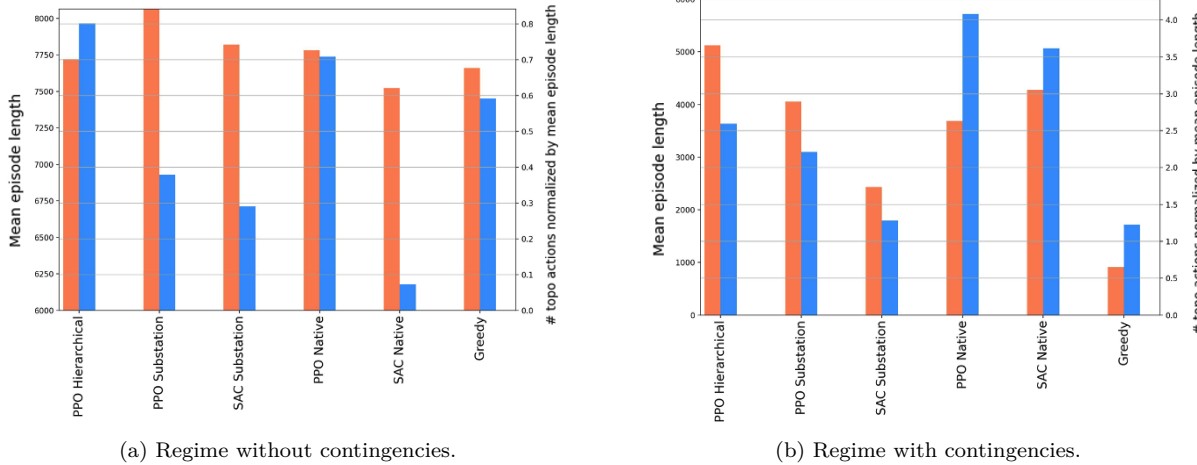

(a) Regime without contingencies.       (b) Regime with contingencies.

Figure 4: Performance of different agents (red bars, left y-axis) as well as the normalized fraction of actions that result in a change of the topology (blue bars, right y-axis) for 200 scenarios in the test set.

More specifically, Figure 4a shows that in case of the less challenging regime without contingencies all agents (including the greedy agent) perform well (note that the left y-axis starts at 6000), solving between 87.5% (SAC Native) to a 100% (PPO Substation) of the test scenarios. The models using PPO show superior performance compared to the respective Native and Substation models that use SAC, though at a cost of increased activity which could be a drawback in a production system. The PPO Hierachical model achieves competitive performance but it is characterized by the highest activity level of all agents. In contrast, SAC Native requires over 11 times less topological changes than PPO Hierarchical still achieving competitive performance. However, SAC Native is the only agent that did not surpass the baseline greedy agent.

In the more challenging regime with contingencies all RL-based models significantly outperform the greedy agent by a factor of 2.6 (SAC Substation) to over 5.6 (PPO Hierarchical) in terms of mean episode length, as shown in Figure 4b. Moreover, all agents change the topology on average 6 times more often as compared to the regime without contingencies. PPO Hierarchical has the best overall performance of all agents with the highest mean episode length and moderate activity level. The PPO Substation performs worse than the SAC Native agent in terms of mean episode length. However, PPO Substation still maintains a relatively low fraction of actions that change the topology while being competitive in performance. The gap between PPO Substation and PPO Native is relatively larger. As reflected by the training curves in Figure 3, the SAC Substation performs significantly worse than all other RL models. PPO Native also achieves relatively good performance with the highest activity level of all agents. It is worth to point that the gap to the theoretical mean episode length upper bound of 8064 is 36% for the best agent, indicating room for improvement.

# 6 Summary and future work

## 6.1 Summary

In this work, we propose a HRL approach to power network topology control. In this approach, the topology control task is decomposed into the following 3-level hierarchy of sub-tasks: (i) the highest-level sub-task is modeled as a rule-based policy over two options, namely, either 'do nothing' or 'propose a topology change'; the second option is further decomposed into (ii) the intermediate-level sub-task 'choose a substation' and (iii) the lowest-level sub-task 'given a substation, choose a substation configuration'. The main advantage of this approach is that it effectively reduces the action space. In this study, the level-i policy is only rule-based, whereas the level-ii policy is learnt via RL. For the level-iii policy we investigate two versions, namely, either also learnt via RL (our advanced 3-level HRL agent) or given by greedy brute-force search (our hybrid 3-level HRL agent).

All agents are trained within the power system environment as given by the IEEE 14-bus network, and two experimental regimes of different difficulty levels: either without contingencies or with contingencies. We thoroughly compare the advanced 3-level HRL agent and the hybrid 3-level HRL agent with each other as well

as with common reference agents, namely, more vanilla RL and greedy brute-force agents that do not enable action space reduction[5]. Moreover, we compare the effects of different RL algorithms, namely, PPO and SAC.

Our experiments show that in the simplified regime without contingencies all agents (including the greedy brute-force search agent) perform well in terms of mean episode length (of course, a brute-force search has significantly longer inference times). The worst agent fails in 13.5% of the test scenarios whereas the best agent, the hybrid 3-level HRL agent using PPO, solves all of the test scenarios. The models trained with SAC proved to be more sensitive to the choice of hyperparameters, and less stable and performant than agents trained with PPO. A reason for that could be that PPO has a loss that explicitly limits the difference between subsequent policies, which might make updates more stable and less likely to diverge than SAC-based updates. In the more challenging regime with contingencies, the greedy agent performs significantly worse than the rest of the agents completing none of the test scenarios. Moreover, the hybrid 3-level HRL approach using the SAC algorithm did not achieve performance competitive with that of the other RL agents. Models trained with PPO again displayed the most consistent training behavior. A key finding is that the advanced 3-level HRL agent (i.e. the agent that employs RL both at the intermediate and the lowest level) outperforms all other agents in the most difficult environment.

The results confirm the hypothesis that grid topology re-configuration is a feasible way to mitigate congestion. When the grid is not pushed close to its limit it is feasible to successfully operate the grid with a simulator and a greedy heuristic given enough time to simulate all topologies. However, the results of the regime with contingencies indicate a need for longer-term anticipation that is exhibited by the RL-based agents. Without tools that suggest anticipatory actions several steps ahead power grid controllers will have to revert to other, likely more expensive types of actions for preventing grid congestion.

### 6.2 Future work

An obvious follow-up of this work is to apply the proposed HRL framework to larger power grids. This study focused on a relatively small power grid of 14 substations to allow for direct comparison with both RL reference agents which do not employ an action space reduction and greedy brute-force search (which additionally employs computation-heavy simulation). We note that in the given format neither the greedy agent nor the native RL agents scale to larger grid sizes due to the exponential increase in available topologies. It is infeasible to simulate all the topologies such that RL exploration in such a huge, flat action space is impractical. That is the reason why most of the top methods in the L2RPN competitions reduced the number of available actions to around 1.5% of the available actions. However, the situation is different for hierarchical agents, where the number of actions for a higher-level policy grows linearly (i.e. more substations to choose from) and the number of actions per substation does not grow with the number of substations. Scaling the hierarchical agent without restricting the action space is a promising avenue of research that could unlock the flexibility of many available grid topologies.

Another interesting avenue of research would be incorporating graph neural network (GNN) based policies into our HRL framework. Early research (Yoon et al., 2020; Liao et al., 2021) has demonstrated that GNNs effectively encode combinatorial and relational input. Usage of GNNs could allow for transfer learning between different grids, potentially improving performance and accessibility for different system operators.

Furthermore, our HRL framework could be improved in several directions. For example, at the highest level a policy-over-options could be learnt or more than two options could be designed. And at the intermediate level temporally extended actions could be added by including sets of substations. Moreover, our results demonstrate that the mean episode reward between the agents is similar (see appendix A) despite significant differences in our main performance criterion (i.e. the mean episode length). Such a phenomenon suggests that the reward signal is not optimal and could be improved. An obvious modification of the reward function could be to include a measure of the robustness of a chosen topology to line outages (see e.g. Lehna et al. (2023)).

Finally, we observe that a bifurcation emerges in hierarchical policy space when training on different random seeds. For agents employing RL at only one level the hierarchical policies (due to different seeds) are unimodally distributed around the mean reward. In contrast, the advanced 3-level HRL agent exhibits dichotomous convergence behavior (leading to a bimodal distribution of hierarchical policies). It could very interesting to understand this behaviour better (e.g. whether it generalizes to other agent architecture and power networks).

---

[5]The reference agents only employ the level-i and level-iii policies and discard the level-ii policy. So strictly speaking these reference agents are 2-level HRL agents. However, these agents are usually not modelled in hierarchical terms (e.g. they are usually described as an MDP instead of an SMDP).

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

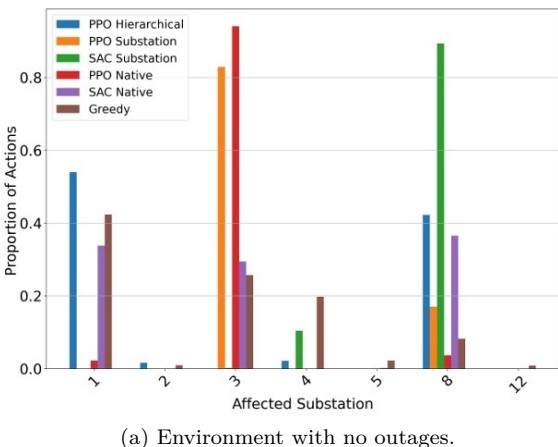 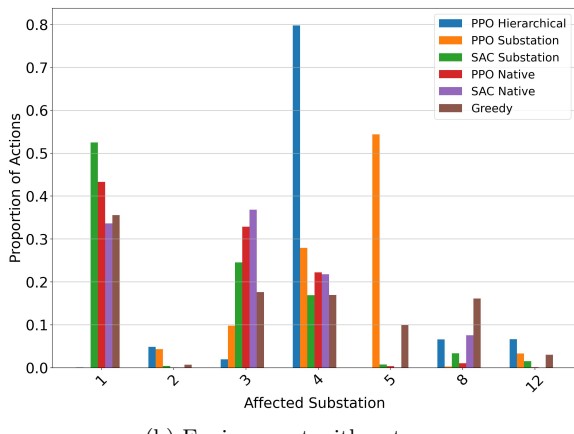

(a) Environment with no outages.          (b) Environment with outages.

Figure 5: The distribution of actions above the activity threshold over the affected substations. The actions that do not change the topology were excluded and the probability distribution was normalized.

Richard S. Sutton, Doina Precup, and Satinder Singh. Between MDPs and semi-MDPs: A framework for temporal abstraction in reinforcement learning. *Artificial Intelligence*, 112(1):181–211, 1999. ISSN 0004-3702. doi: https://doi.org/10.1016/S0004-3702(99)00052-1. URL https://www.sciencedirect.com/science/article/pii/S0004370299000521.

R.S. Sutton and A.G. Barto. *Reinforcement Learning: An Introduction*. MIT Press, 2018.

Leon Thurner, Alexander Scheidler, Florian Schäfer, Jan-Hendrik Menke, Julian Dollichon, Friederike Meier, Steffen Meinecke, and Martin Braun. pandapower—an open-source python tool for convenient modeling, analysis, and optimization of electric power systems. *IEEE Transactions on Power Systems*, 33(6):6510–6521, 2018.

Jan Viebahn, Matija Naglic, Antoine Marot, Benjamin Donnot, and Simon H. Tindemans. Potential and challenges of AI-powered decision support for short-term system operations. In *CIGRE Session 2022*, 2022.

Deunsol Yoon, Sunghoon Hong, Byung-Jun Lee, and Kee-Eung Kim. Winning the l2rpn challenge: Power grid management via semi-markov afterstate actor-critic. In *International Conference on Learning Representations*, 2020.

Chao Yu, Jiming Liu, Shamim Nemati, and Guosheng Yin. Reinforcement learning in healthcare: A survey. *ACM Comput. Surv.*, 55(1), 2021. ISSN 0360-0300. doi: 10.1145/3477600. URL https://doi.org/10.1145/3477600.

Bo Zhou, Hongsheng Zeng, Yuecheng Liu, Kejiao Li, Fan Wang, and Hao Tian. Action set based policy optimization for safe power grid management. In Yuxiao Dong, Nicolas Kourtellis, Barbara Hammer, and Jose A. Lozano (eds.), *Machine Learning and Knowledge Discovery in Databases. Applied Data Science Track*, pp. 168–181, Cham, 2021. Springer International Publishing.

## A    Detailed performance analysis

Figure 5 illustrates the distribution of affected substations for all agents in the unseen test scenarios. The agents' strategies exhibit significant variations. In the environment without outages (Fig. 5a), the greedy agent displays the most diverse distribution, intervening at least once in each substation. Conversely, the RL-based agents primarily take actions on substations 1, 3, and 8. The presence of outages (Fig. 5b) induces more diverse behavior across all agents, resulting in a more balanced distribution, except for the PPO Hierarchical agent, which focuses almost 80% of its actions on substation 4.

We further examine various metrics in Table 2 (no contingencies) and Table 3 (with contingencies) to better understand the agents' performance and strategies. The first and third rows reiterate the findings shown in Figure 4: In the regime without contingencies (Fig. 4a) agents exhibit significant differences in the number of actions changing the topology but not as much in performance measured by the mean episode length. In contrast, in the regime with contingencies (Fig. 4b) both the mean episode length and the amount of topological actions vary significantly between agents. In particular, the agents with the lowest mean episode length (SAC

| | PPO Hierarchical | PPO Native | PPO Substation | SAC Native | SAC Substation | Greedy |
|---|---|---|---|---|---|---|
| Mean episode length | 7720.16 (7843.9505) | 7781.99 (7917.48) | 8064 (8045.63) | 7522.95 (7736.74) | 7821.42 (7724.33) | 7659.32 (7809.46) |
| Mean normalized reward | 6829.83 | 6870.43 | 6874.94 | 6859.10 | 6860.19 | 6831.44 |
| # of topo changes | 6189 | 5514 | 3056 | 550 | 2276 | 4534 |
| # unsolved scenarios | 20 | 15 | 0 | 27 | 15 | 23 |
| # of unique topologies | 59 | 57 | 62 | 8 | 65 | 1916 |
| Mean topo depth | 2.68 | 4.38 | 3.58 | 4.00 | 2.39 | 5.76 |
| St. dev. of topo depth | 0.59 | 1.21 | 0.79 | 1.22 | 0.83 | 1.72 |
| # unique sequences | 443 | 257 | 47 | 1 | 47 | 242 |
| Mean sequence length | 3.17 | 3.22 | 2 | nan | 2.15 | 2 |
| St. dev. of sequence length | 1.68 | 1.22 | 0 | nan | 0.36 | 0 |
| Mean sequence repeatability | 2.70 | 5.32 | 1.09 | 1 | 4.02 | 1.05 |
| St. dev. of sequence repeatability | 8.45 | 17.57 | 0.28 | 0 | 5.75 | 0.54 |

Table 2: Summary statistics for different agents on the test set in the environment without outages. The mean episode length for combined train, validation and test set is given in parenthesis. The abbreviations are: # - number, st. dev. - standard deviation, topo - topological.

| | PPO Hierarchical | PPO Substation | PPO Native | SAC Native | SAC Substation | Greedy |
|---|---|---|---|---|---|---|
| Mean episode length | 5124.69 (5147.66) | 4052.91 (4169.29) | 3688.20 (4054.57) | 4277.88 (4449.58) | 2429.20 (2698.70) | 911.25 (950.16) |
| Mean normalized reward | 6948.04 | 6927.15 | 6940.01 | 6948.99 | 6922.36 | 6916.64 |
| # of topo changes | 13296 | 8959 | 15042 | 15457 | 3116 | 1116 |
| # unsolved scenarios | 132 | 156 | 174 | 152 | 195 | 200 |
| # of unique topologies | 41 | 141 | 169 | 83 | 219 | 308 |
| Mean topo depth | 4.81 | 4.16 | 4.49 | 4.05 | 4.33 | 3.40 |
| St. dev. of topo depth | 0.68 | 0.93 | 0.87 | 0.79 | 1.22 | 1.69 |
| # unique sequences | 404 | 273 | 1173 | 854 | 132 | 71 |
| Mean sequence length | 3.54 | 2.40 | 3.36 | 4.37 | 2.30 | 2 |
| St. dev. of sequence length | 2.33 | 0.80 | 1.93 | 3.78 | 0.59 | 0 |
| Mean sequence repeatability | 6.31 | 6.92 | 2.17 | 3.44 | 1.73 | 1.37 |
| St. dev. of sequence repeatability | 31.10 | 22.88 | 5.83 | 12.03 | 2.40 | 1.55 |

Table 3: Summary statistics for different agents on the test set in the environment with outages. The mean episode length for combined train, validation and test set is given in parenthesis. The abbreviations are: # - number, st. dev. - standard deviation, topo - topological.

Substation and Greedy) also exhibit the smallest amount of topological actions which suggests that contingencies necessitate more topologcial actions for an agent to be successful (as expected). Moreover, the fourth row shows the number of unsolved test scenarios. Obviously, this measure is anti-correlated with the mean episode length.

The second row shows mean episode reward (i.e. the total reward of an episode divided by the episode length) averaged across all test scenarios and multiplied by 8064 (the total number of timesteps of a scenario) to make differences more visible. This measure is similar among all agents both in the regime without contingencies (Fig. 4a) and in the regime with contingencies (Fig. 4b). Such a phenomenon suggests that the reward signal is not optimal and, hence, the definition of the reward function could be improved.

The remaining rows list measures related to the topological properties and the sequence character of the agents' actions (we define a sequence as more than one action that changes the topology executed in consecutive time steps). Interestingly, the algorithms that use RL induce staggeringly fewer topologies than the greedy agent (see fifth row). Moreover, the sixth row indicates that the Greedy agent induces very high topological depths (i.e. topologies that are far away from the default topology where in each substation all elements are connected to the same busbar) in the regime without contingencies whereas in the regime with contingencies the greedy shows lower mean topological depth than the RL agents (which probably is simply a consequence of the Greedy agent's much shorter episode lengths).

Finally, all agents can employ a large amount of unique sequences. However, the Greedy agent is limited to sequences of length 2 in both regimes whereas the RL-based agents are able to come up with longer sequences and higher repeatability thereof - as expected for a sequential decision-making approach like RL.

## B  Hyperparameters

For all models we use the discount factor $\gamma = 0.99$ and the activity threshold $\rho = 0.95$. Below this threshold, a do-nothing action is executed and the policy is queried for the action above the threshold. Each model has 3 hidden layers with ReLU activations Fukushima & Miyake (1982), each with 256 units. The hidden layers are preceded by a projection layer that embeds the observation into a 256-dimensional vector. The output layer, combined with softmax normalization, projects the last hidden embedding into a multinomial probability distribution over (1) substations (8 units) or (2) topologies (106 units). Actor and critic networks employ the same learning rate.

We conduct grid search for each model, algorithm, and training combination. We discover that the parameters of the models trained on a flat, native action space transfer well (i.e., stable and rapid training) to hybrid models and less effectively to the full hierarchical model.

### B.1  PPO

The parameters used for PPO models are shown in Table 4. Note that the hierarchical agent consists of two PPO policies. Both policies use the same hyperparameters. In course of searching for optimal parameters, the most crucial parameters were the batch size and the number of SGD iterations performed on each batch. We found that a larger batch size combined with a smaller number of SGD iterations yielded more stable training and higher expected rewards.

Notably, Rllib's PPO implementation uses not only the clipped surrogate objective as described by Schulman et al. (2017) but also adds (1) a fixed KL penalty and (2) the entropy term. The first modification was proposed in the PPO paper Schulman et al. (2017) as an alternative to the clipping objective and the second modification was borrowed from maximum entropy reinforcement learning algorithms such as SAC. The objective with these modifications becomes:

$$J_\pi(\phi) = \mathbb{E}_t \left[ \min \left( r_t(\theta)\hat{A}_t, \text{clip}\left( r_t(\theta), 1 - \epsilon, 1 + \epsilon \right) \hat{A}_t \right) - \beta \text{KL}\left[ \pi_{\theta_{\text{old}}}, \pi_\theta \right] + \alpha H \left( \pi \left( . \mid s_t \right) \right) \right], \tag{1}$$

A small KL coefficient proved beneficial for stabilizing training. However, we empirically found that adding the entropy term does not result in better performance for PPO agents except for the scenario with outages for PPO Native and PPO Hierarchical.

Table 4: Hyperparameters for agents trained with PPO.

| Experiment | No Outages | | | Outages | | |
|---|---|---|---|---|---|---|
| Model | Native | Substation | Hierarchical | Native | Substation | Hierarchical |
| LR | 1e-4 | 1e-4 | 5e-4 | 1e-4 | 1e-4 | 5e-4 |
| KL coeff | 0.2 | 0.2 | 0.3 | 0.2 | 0.2 | 0.3 |
| Clip param | 0.3 | 0.3 | 0.5 | 0.3 | 0.3 | 0.5 |
| Entropy coeff | 0 | 0 | 0 | 0.01 | 0 | 0.025 |
| SGD iters | 5 | 5 | 15 | 15 | 15 | 8 |
| Minibatch size | 256 | 256 | 256 | 256 | 256 | 256 |
| Batch size | 1024 | 1024 | 1024 | 1024 | 1024 | 1024 |

For further details on PPO parameters please consult RLlib's PPO documentation.

### B.2  SAC

The parameters used for training SAC models are shown in Table 5. The rewards obtained from the environment are scaled by a factor of 3. As discussed in the paper that introduced SAC Haarnoja et al. (2018a), the reward scale controls the stochasticity of the optimal control. Without scaling the policy fails to exploit the reward signal and performs significantly worse. We also found that keeping the entropy fixed during training resulted in better performance than adaptively changing it as proposed by Haarnoja et al. (2018b). All models use experience replay Schaul et al. (2015) with the same parameters: $\alpha = 0.6, \beta = 0.4$.

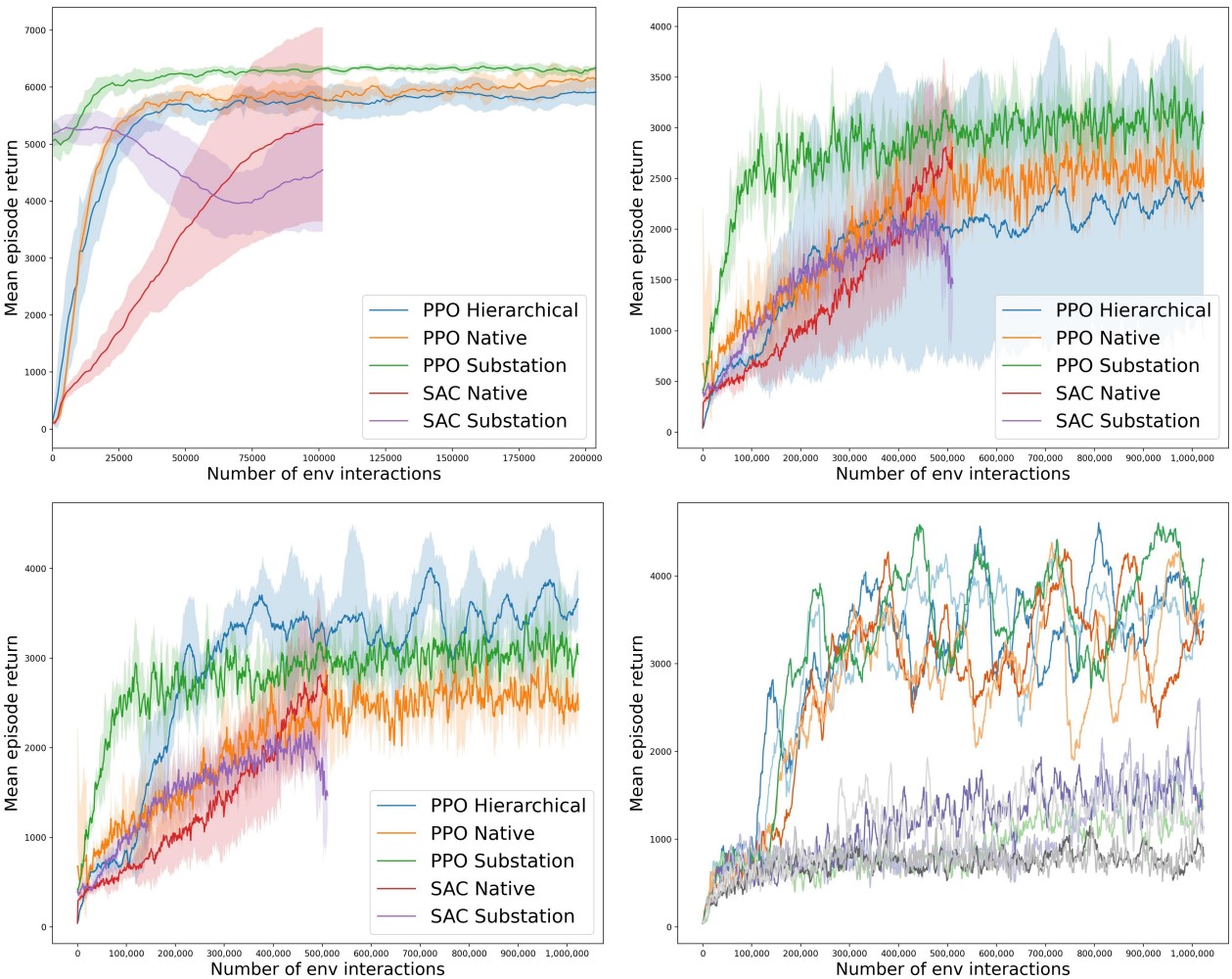

Figure 6: Training curves of the different RL agents. Same as Fig. 3 except that the curves for the PPO agents are twice as long.

Table 5: Hyperparameters for agents trained with SAC.

| Experiment | No Outages | | Outages | |
|---|---|---|---|---|
| Model | Native | Substation | Native | Substation |
| LR | 1e-4 | 1e-4 | 1e-4 | 1e-4 |
| $\tau$ | 5e-3 | 5e-4 | 5e-3 | 5e-4 |
| Entropy coefficient | 0.05 | 0 | 0.01 | 0 |
| Target network update freq | 100 | 10 | 100 | 10 |
| Batch size | 512 | 512 | 512 | 512 |

For further details on SAC parameters please consult RLlib's SAC documentation.

## C  Extended training curves

In order to provide a fair comparison between the training curves of the PPO models and SAC models we chose to depict the training progress in terms of number of environment interactions, as shown in Fig. 3. However, as summarized in appendix B, hyperparameter tuning lead to different preferred batch sizes for PPO (see Tab. 4) and SAC (see Tab. 5), respectively. Since we computed the same amount of training batches for both PPO models and SAC models and the batch size of PPO models is larger we actually have longer training curves available for the PPO models. Figure 6 shows the same panels as Fig. 3 but includes the extended training curves of the PPO models. The main conclusion from Fig. 6 is the that training of the PPO models is indeed converged after 500k environment interactions (i.e. where Fig. 3 ended).

