# OpenReview forum: "Hierarchical Reinforcement Learning for Power Network Topology Control"
_TMLR — Rejected by TMLR_

### Review · Reviewer_VBby · 2023-05-26

**Summary Of Contributions:**

In this paper, hierarchical reinforcement learning is investigated for simulations of power network topology control. The task is divided into three hierarchical levels, where the middle and partly also the lowest level are optimized by RL. As RL algorithms PPO and SAC are investigated.

**Audience:**

Yes

**Claims And Evidence:**

No

**Requested Changes:**

I think one possibility would be to send the paper to a more application-oriented journal or a journal on power grid control.

The other possibility would be to make the studies much more extensive, and among other things to investigate a systematic comparison of HRL and RL for several different sized problems, ideally showing that HRL shows superior performance above a certain problem size.

Minor:\
"a RL" -> "an RL"

"appendix A" -> "Appendix A"

"Hierachical" -> "Hierarchical"

"markov" -> "Markov"


**Strengths And Weaknesses:**

**Weaknesses**
* The advantage of HRL compared to RL is not made clear enough.
* It remains unclear what the scientific contribution is, the paper reads more like a project report.
* The observations presented are not sufficiently analyzed to draw conclusions that can be generalized.

---

> ### Author Response · Authors · 2023-07-04
> **Response to Review by Reviewer VBby**
>
> We thank Reviewer VBby for providing a review of our submission and for the suggestions to improve it. We reply to the concerns raised point-by-point below:
>
> - Regarding the comparison of HRL and RL for several differently sized problems: We completely agree that applying the provided HRL framework to larger power grids is an obvious next step. We also propose this step in the first paragraph of Section 6.2 on future work. However, we consider this as a study on its own. The current study is focused on (see also the next bullet point) providing a novel formulation of the power network control problem as a HRL problem and demonstrating the potential of HRL via a clean comparison with other approaches. As pointed out in the first paragraph of Section 6.2, the greedy approach and the plain RL approach are not directly scalable to larger grids but a significant amount of additional heuristics is necessary (see e.g. [1]). Consequently, in a follow-up study focused on larger grids a detailed analysis of the different techniques employed in different approaches is necessary in order to make the comparison fair and meaningful. In the current study we could keep it clean by focusing on the hierarchical structure only (see e.g. the right panel of our Figure 2).
>
>     $[1]$ Lehna, M. et al., Managing power grids through topology actions: A comparative study between advanced rule-based and reinforcement learning agents, Energy and AI, Volume 14, 2023
>
> - Regarding the key contributions: We agree that perhaps we did not make our contributions explicit enough. We consider our contributions to be:
>
>     1. We propose a novel formulation of the power network topology control problem as a hierarchical reinforcement learning (HRL) problem, providing an effective decomposition of the action space.
>     2. We offer a comprehensive comparison between conventional RL, expert control, hybrid, and fully hierarchical approaches. Our evaluation is conducted on a single-scale power network where such comparisons are computationally feasible. The former two approaches are not scalable to larger grids due to the combinatorial explosion of the action space.
>     3. We demonstrate that the fully hierarchical approach outperforms the other approaches in the more challenging and realistic environment (i.e., power systems with contingencies), providing concrete evidence of the potential of this approach.
>
>     We will update the paper to explicitly state these contributions.
>
> - Regarding the advantage of HRL to RL and the depth of the analysis:
> We are happy to revise the paper in order to better point out the key advantage of HRL in our context, namely, that HRL has the potential to handle large action spaces more effectively by decomposing the action space and thus allowing for more effective learning. However, we also like to mention that the objective of our study is not to praise HRL, but to provide a systematic study that might reveal disadvantages, advantages, or a lack of substantive effects. More precisely, we would like to point out that the power network control problem is a real-world sequential decision-making problem which is of both enormous complexity and high social relevance in times of the energy transition. But at the same time, the assessment (and subsequently further development) of RL approaches for this complex real-world learning task is still in its infancy (as mentioned at the beginning of Section 1.1.1). Our paper is the first study that is focused on formalizing the power network control problem as a HRL problem and systematically assesses different HRL approaches across different difficulty levels of the task (with or without contingencies) and various performance measures. And even though making HRL work is not self-evident we provide a successful approach with RL on two levels (i.e. with interacting RL policies).

---

> > ### Comment · Reviewer_VBby · 2023-07-25
> > **Acknowledgement of rebuttal**
> >
> > The authors' responses strengthen my opinion that the paper is better suited for a journal that is more application-oriented, where `a novel formulation of the power network topology control problem` is valued as a gain in itself. I consider the contribution to *machine learning research* to be too small for publication in TMLR.

---

> > > ### Author Response · Authors · 2023-07-26
> > > **Response to reply by Reviewer VBby**
> > >
> > > We thank Reviewer VBby for the reply. We would like to point out that TMLR provides a detailed explanation of its scope (i.e. what is considered 'machine learning research' in the context of TMLR) and evaluation criteria: https://jmlr.org/tmlr/editorial-policies.html. In the exchange with Reviewer 26Bd we explain in detail to which scope categories our paper contributes.
> > >
> > > Moreover, we would like to point out that Reviewer VBby is only citing the first half of our first contribution (where the application is mentioned) and ignores the other 2.5 of our contributions. We think that in order to draw conclusions Reviewer VBby should also criticize in detail what we mention as our 2. and 3. contributions.

---

### Review · Reviewer_26Bd · 2023-06-02

**Summary Of Contributions:**

This paper applies hierarchical RL to control a simulation of power networks. HRL reduces the size of the combinatorial action space, and is shown to be marginally better than the non-hierarchical approach.

**Audience:**

No

**Broader Impact Concerns:**

None.

**Claims And Evidence:**

No

**Requested Changes:**

This work should be submitted to a venue more suitable for the application of interest.

**Strengths And Weaknesses:**

The main strength is a detailed and clearly reported experimental study.

Some weaknesses:
- I am not sure that this paper satisfies any of the criteria in the call for papers by TMLR. This is an experimental study of known methods, but I do not think the study provides new generalizable insights about the design and behavior of these methods. Comparing training stability of PPO and SAC (used as the base RL in the hierarchy) is orthogonal to the understanding of hierarchical RL methods. The benefits of hierarchical RL over non-hierarchical methods is already well known, and this demonstration in the particular application of power network control has not added new findings.
- The benefits of using two levels of RL ("Hierarchical RL") versus only one level ("RL substation") is not clear. In fact, "RL substation" is the best in average performance, which shows that the factorization of action space helps the most and it doesn't matter whether RL or a greedy strategy is used at the primitive level.
- Lack of comparison to the state of the art. The authors describe competitions held at previous conferences in this application area, but do not compare to any of them.
- Claims are not stated precisely enough to let readers see what exactly is the new knowledge contained in this work. It is for this reason that I indicate "no" for claims and evidence below.

The authors should also note that the case where two separate RL agents are applied at the intermediate and lowest levels is actually a special case of a Stackelberg game where both leader and follower get the same reward.

---

> ### Author Response · Authors · 2023-07-03
> **Response to Review by Reviewer 26Bd (part 1)**
>
> We thank Reviewer 26Bd for reviewing our submission, and for noting both strengths and points of improvement for the paper. We reply to the concerns raised point-by-point below:
>
> - Regarding being within the scope of TMLR: We thank the reviewer for voicing this concern. First of all, we would like to point out that the power network control problem is a real-world sequential decision-making problem which is of both enormous complexity and high social relevance in times of the energy transition.
> At the same time, the assessment (and subsequently further development) of RL approaches for this complex real-world learning task is still in its infancy (as mentioned at the beginning of Section 1.1.1). Our paper is the first study that is focused on formalizing the power network control problem as a HRL problem and systematically assesses different HRL approaches. In this regard, we would like to point out that TMLR explicitly considers "accounts of applications of existing techniques that shed light on the strengths and weaknesses of the methods" and "formalization of new learning tasks (e.g., in the context of new applications)" in scope. We consider our paper falling in these categories.
>
> - Regarding the benefits of HRL to RL: We agree that many papers on HRL have already been written, however, we don't quite agree that the benefits are that clear. Results on HRL, expecially in the single task setting, tend to be quite mixed unless strong prior structure is given. Furthermore, much of the prior work has looked at HRL more as method to provide temporal or spatial abstraction, and less as a tool to handle a large and complex action space. In other words, we don't consider making HRL work as being straightforward. Nevertheless, we provide a successful approach including even RL on two levels (i.e. interacting RL policies).
>
> - Regarding the benefits of using two levels of RL ("Hierarchical RL") versus only one level ("RL substation"): It is true that for the simpler and less realistic regime (i.e. power system environment without contingencies, Section 4.1.1) PPO Substation achieves the best performance. However, in this less challenging regime all agents (including the greedy agent) perform well. This is pointed out at the beginning of Section 5.2. Note that the left y-axis in the left panel of Figure 4 only starts at 6000, and also in Table 2 in the row of the mean episode length all values are rather close. In contrast, in the more challenging and realistic regime (i.e. power system environment with contingencies, Section 4.1.2) the situation is significantly different (as also described in Section 5.2). In this case PPO Hierarchical has the best overall performance of all agents as is shown in the right panel of Figure 4 and in the first row of Table 3.
>
> - We completely agree that much more systematic benchmarking of RL approaches applied to power network control is necessary. One problem with the currently available approaches is that these are often partly tweaked for specific competition setups. For example, some approaches employ specific hard-coded sets of topologies which have been identified by the participants in an exploratory phase preceding the actual model training phase. We prefer a clean approach by employing only generic techniques without specific tweaks (see e.g. the action space restriction in Section 2.4 which works for any network). Nevertheless, the PPO Native agent can be considered as a state-of-the-art reference since it has been employed in high-ranking competition submissions (see 2nd place in [1]) and subsequent research [2, 3] (important to note that a significant amount of extra heuristics is necessary to make it work on larger networks).
>
>    $[1]$ Marot, A. et al., Learning to run a Power Network Challenge: a Retrospective Analysis, Proceedings of the NeurIPS 2020 Competition and Demonstration Track, PMLR 133:112-132, 2021
>
>    $[2]$ Lehna, M. et al., Managing power grids through topology actions: A comparative study between advanced rule-based and reinforcement learning agents, Energy and AI, Volume 14, 2023
>
>    $[3]$ Chauhan, A. et al., PowRL: A Reinforcement Learning Framework for Robust Management of Power Networks, Proceedings of the AAAI Conference on Artificial Intelligence, 37(12), 2023

---

> > ### Author Response · Authors · 2023-07-03
> > **Response to Review by Reviewer 26Bd (part 2)**
> >
> > - We agree that perhaps we did not make our contributions explicit enough. We consider our contributions to be:
> >
> >     1. We propose a novel formulation of the power network topology control problem as a hierarchical reinforcement learning (HRL) problem, providing an effective decomposition of the action space.
> >
> >     2. We offer a comprehensive comparison between conventional RL, expert control, hybrid, and fully hierarchical approaches. Our evaluation is conducted on a single-scale power network where such comparisons are computationally feasible. The former two approaches are not scalable to larger grids due to the combinatorial explosion of the action space.
> >
> >     3. We demonstrate that the fully hierarchical approach outperforms the other approaches in the more challenging and realistic environment (i.e., power systems with contingencies), providing concrete evidence of the potential of this approach.
> >
> >     We will update the paper to explicitly state these contributions.
> >
> > - Regarding the link to Stackelberg games: We thank the reviewer for this note. We are happy to point out the connection in the paper.

---

> > ### Comment · Reviewer_26Bd · 2023-07-13
> > **Acknowledgement of rebuttal, some concerns remain.**
> >
> > I acknowledge the explanations that 1) the hierarchical action space factorization in this work differs from prior HRL methods focused on temporal abstraction; 2) there is one result in Figure 4 that shows HRL has strengths over the non-hierarchical methods; 3) the PPO agent can be considered a surrogate of previous methods submitted to the competition on this topic.
> >
> > These are the remaining areas where the paper can be improved:
> > - Comparison to SOTA: the clarifications in the rebuttal should be inserted in some form into the paper, otherwise readers will have the same questions about how to situate these results in the context of those competition results.
> > - Relevance to TMLR:
> >   - If the authors intended this paper to contribute to the category of "accounts of applications of existing techniques that shed light on the strengths and weaknesses of the methods", then it should be necessary to provide 1) explanations for any observed strengths and weaknesses, and 2) concrete evidence that back up these explanations. Using a known approach for the first time on an application problem is not enough to satisfy this requirement. Currently, it appears that the main relevant finding is that comparison of HRL versus non-hierarchical methods is inconclusive in the case without contingencies, while one instance of HRL does better in the case with contingencies. This is an observation, which by itself does not satisfy the criteria of relevance. It should be supplemented with evidence-based explanations, e.g. what is it about the nature of the two problem cases that caused the difference in performances.
> >   - I disagree that this paper contributes to the category of "formalization of new learning tasks (e.g., in the context of new applications)," since there is prior work that formulated the application as an MDP. The action space factorization may be new in this application, but the learning task is not.
> > - Clarity and precision of positioning: Referring again to the result that "PPO substation" has strong performance (best in case without contingency, 2nd best in case with contingency), it appears that the action space factorization matters the most. This factorization technique by itself should not be called hierarchical RL if RL is applied only at one level, since HRL is widely understood in the literature as referring to learning at multiple levels or temporal abstractions (i.e. SMDPs), see the survey [1].
> >
> > [1] Pateria et al. Hierarchical reinforcement learning: A comprehensive survey. ACM Computing Surveys 2021.

---

> > > ### Author Response · Authors · 2023-07-15
> > > **Response to the reply by Reviewer 26Bd**
> > >
> > > We thank Reviewer 26Bd for acknowledging part of our explanations, and for suggesting further clarifications. We reply to the points raised point-by-point below:
> > >
> > > Revising the paper:
> > > - We totally agree. We will include clarifications in the paper as soon as these are agreed by the reviewers. We will submit a revised version in the upcoming week that includes the clarifications regarding SOTA.
> > >
> > > Relevance to TMLR:
> > > - We are happy to provide more explanations regarding the two experimental cases (i.e. without contingencies vs with contingencies) and the related performances. In the case without contingencies (which is considered in several previous studies) sequential decision making turns out to be less important. This is visible in Table 2 which shows that the greedy agent (i.e. no time horizon is taken into account) achieves a similar mean episode length (see 1st row in Table 2) as the RL agents. For all agents the mean episode length is close to the optimum of 8064 steps. The biggest relative difference between agents in terms of mean episode length is only about 7% for PPO Substation vs SAC Native. The greedy agent achieves this with a mean sequence length (see 9th row in Table 2) of only 2 (the same for the best RL agent, PPO Substation, which also employs greedy behavior). The reason for this behavior is that in the case without contingencies all elements of the network remain in service such that an agent ‘only’ needs to respond to changes in the load flow due to the evolution of the power injections. The evolution of the power injections is smooth and localized (at specific substations) such that targeted greedy actions often can resolve the congestion which is signaled early by surpassing the activity threshold.
> > >
> > >     The situation is very different in the case with contingencies! In this case additionally random outages at possibly very different locations in the network can induce congestion. This requires much more suboptimal short-term (i.e. non-greedy) behavior in order to be prepared for unpredictable and spatially diverse outages. In other words, an agent needs to prepare for (i.e. choose topologies that are robust to) several forms of congestion at the same time (greedy optimization can only focus on the current situation). Consequently, the greedy agent performs much worse in this case as can be seen in the first row of Table 3. Moreover, also between RL agents the mean episode length is much more diverse with the biggest relative difference being about 52% (PPO Hierarchical vs SAC Substation), and the smallest relative difference still being about 17% (PPO Hierarchical vs SAC Native). Hence, in this case PPO Hierarchical starts to shine. In particular, applying RL also at the lowest hierarchical level (i.e. the choice of the substation configuration) is beneficial as the comparison with PPO Substation shows. The reason is that greedy behavior can only help after the contingency and resulting congestion happened (quickly finding a solution for the current situation) which, however, often can be too late. It is better to anticipate/prepare for outages (i.e. by choosing robust topologies) which can only be done by non-greedy behavior. Hence, the low-level control  is also sensitive to long-term return. Note that the mean sequence length increased to about 3.5 for PPO Hierarchical.
> > >
> > > - We agree that the MDP formulation of the power network control problem has been given in previous publications. However, we would like to point out again that this is this first study which employs a dedicated HRL perspective on the problem which implies (amongst other aspects) formulating the problem as an SMDP instead of an MDP (see the Sections 1.1.2, 3.1.1, and footnote 5 in Section 6.1). More precisely, in Section 3.1.1 we re-formulate a simple high-level rule (practically employed in many previous studies) in the options framework. In this way we make the SMDP character of the problem clearer and suggest new approaches for future research like designing more options or learning a policy-over-options (see Sections 3.1.1 and 6.2).
> > >
> > > Regarding HRL terminology:
> > > - As explained in detail in the previous point, we consider PPO Hierarchical to be the most performant approach in our study. (Note that PPO Substation is only third place in the case with contingencies.)

---

> > > > ### Author Response · Authors · 2023-07-19
> > > > **Paper revised**
> > > >
> > > > We made SOTA regarding RL applied to power network control more explicit. For that the fourth paragraph of Section 1.1.1 and the first paragraph of Section 3.2 are extended/rewritten.

---

### Review · Reviewer_KGUv · 2023-06-24

**Summary Of Contributions:**

The authors aim to handle the curse of dimensionality issue that arise in power network control, where possible actions to take for each step scale exponentially with the number of substations. The authors propose to use a hierarchical RL approach, where (i) an intermediate-level agent controls the substation to modify at each step, (ii) and a lower-level agent controls the configuration for the selected substation, and (iii) a high-level agent decides whether to make an adjustment or not. The authors then conducted extensive experiments on the 14-Bus system benchmark and tested various combinations of algorithms at each level of control. For the evaluation, the authors innovated the contingencies to make the simulation environment closer to reality. In conclusion, the authors find that RL in general performs better than brute force methods, typically in environments with contingencies.

**Audience:**

Yes

**Broader Impact Concerns:**

N.A.

**Claims And Evidence:**

Yes

**Requested Changes:**

See questions and comments above.

**Strengths And Weaknesses:**

## Strength:

1. The proposed approach is novel to me and intuitive in resolving the curse of dimensionality that arise in the poser network control problem.

2. The authors conducted comprehensive experiments on their proposed methods with various hierarchical designs. The experiments are evaluated in environments with contingencies, which is more realistic and makes the experimental results more compelling. I think the proposed environment with contingencies also has the potential to serve as a benchmark for future studies.


##  Weakness & Questions:
Regarding reward margin:
1. The reward function is convex in margin (and is different from the one adopted in [1], which is concave in margin), so maximizing it will not lead to evenly distributed power loads for all lines. On the contrary, it will encourage uneven loading of energy (i.e., encourage the situation where few lines have large margin). For example, if we have two lines with thermal limits 1, then (i) having one line with 1 amp and another line with 0 amp leads to total reward R=1, whereas (ii) both lines with 1/2 amp leads to total reward R=1/4.


2. The reward is not penalizing overloaded lines. Is there a way to avoid the policy converging to a situation where a few lines are heavily overloaded so that the rest of the lines have larger margins?

Regarding the hierarchical design:
1. In typical hierarchical RL setup the lower level will execute till an termination state. Does the authors adopt similar algorithm design and what is the termination state in this hierarchy RL setup? From the algorithmic description in section 3.2.1, the proposed method feels more like a decoupling of policy (in regular RL setup) into direct product of selecting the substation and selection configuration for the substation.

2. The authors mention that the value network has shared parameters. How is this implemented? If my understanding of the action mask is correct, then the implementation by the authors still needs the output dimension of the value network to be the same as the entire action space. In that case, I feel that the hierarchical method proposed does not effectively reduce the curse of dimensionality from actions in RL.

3. The controller is adjusting only one substation at a time in the intermediate level. Is this setup sufficient to cover all possible actions?

Other comments & minor suggestions:

1. Given that low-level brute force control is leading to a performance gap between RL approach, do we anticipate that the low-level control is sensitive to long-term return instead of greedy at the next step? How does this scenario fit into the substation configuration problem that the low-level control aims to solve? I think it is worth more discussion regarding this performance gap.

2. How does the proposed method compares to previous approaches, e.g., [2] cited by the authors, which also employes a hierarchical design (especially in the proposed environment with contingencies)? More comprehensive comparison can help readers better place the proposed work among existing lines of research.


[1] Antoine Marot et al., Learning to run a power network challenge for training topology controllers. 2019.

[2] Yoon et al., WINNING THE L2RPN CHALLENGE: POWER GRID MANAGEMENT VIA SEMI-MARKOV AFTERSTATE ACTOR-CRITIC. 2021.

---

> ### Author Response · Authors · 2023-07-01
> **Response to Review by Reviewer KGUv (part 1)**
>
> We thank Reviewer KGUv for her/his response, and for pointing out several strengths of our paper. Moreover, we appreciate the detailed questions and constructive suggestions. In the following, we respond to each question raised:
>
> Regarding reward margin:
> 1. We agree with Reviewer KGUv that the topic of suitable reward functions in the context of power network control is an interesting question that requires further research. Accordingly, we point towards possible improvements of the reward function in Section 6.2 on future work. However, this topic is not the focus of our paper and, hence, we employed the most commonly used reward function, namely, the L2RPN reward (https://grid2op.readthedocs.io/en/latest/reward.html#grid2op.Reward.L2RPNReward). This is actually the same reward function as defined in Reviewer KGUv's reference [1]. The only difference is that we additionally scale the reward by the (constant) number of lines such that the reward always falls between 0 and 1. Consequently, our equations in Section 2.5 are not entirely correct and we will correct that. Thank you very much for pointing that out!
>
> 2. We agree with Reviewer KGUv that it could be helpful if the reward function penalizes overloaded lines. Currently, penalizing overloaded lines is not done via the reward function but via soft constraints in the power system environment (see Section 2.3).  This rules out completely pathological behavior.
>
> Regarding the hierarchical design:
> 1. As the Reviewer KGUv correctly points out, we do not employ temporally extended actions at the intermediate level (see also Section 3.1.2). Consequently, the termination state for both the intermediate level and the lowest level is given via the highest level option. Indeed, in the current setup the intermediate level and the lowest level can be understood as a factorization of the decision making process into multiple policies operating at different levels of abstraction. In the current setup the intermediate level solely represents a form of network abstraction instead of temporal abstraction. In future work the intermediate level approach could be extended to also include temporally extended actions (this is also mentioned in Section 6.2). We note that there are other papers that choose new high level actions at each time step, for example, the HIRO approach (https://proceedings.neurips.cc/paper/2018/hash/e6384711491713d29bc63fc5eeb5ba4f-Abstract.html).
>
> 2. We are happy to clarify details of our implementation. As delineated in Appendix B, our architecture initiates with a projection layer which precedes the hidden layers and which is responsible for embedding the observation into a fixed-size vector. However, the dimensions of the observations for the intermediate policy and the lowest-level policy are different: the latter features an appended one-hot vector encoding of the chosen substation. After embedding, these observations yield fixed-size vectors allowing for parameter sharing across both levels for the value function. But we note that the value network's output is scalar, thus rendering its output dimension as 1.
>
>     In contrast, the output dimension of the actor policy at the lowest level corresponds to the size of the primitive action space. Specifically, employing a Multi-Layer Perceptron (MLP)-based policy necessitates the execution of a softmax operation over the masked logits of the entire action space. This process introduces indeed computational overhead which can magnify when scaled to significantly larger power grids. This problem (i.e. reducing the size of the primitive action space) presents an intriguing avenue for future research. The bottleneck could potentially be mitigated by adopting a Graph Neural Network (GNN) policy network (as also mentioned in Section 6.2).
>
>     Despite this challenge, our model effectively handles learning within the combinatorial action space. The action space at the lowest level policy is constrained by the selected substation which forms part of the observation for the lowest-level policy. As a result, the policy must acquire a conditional probability distribution over actions appropriate for a specific substation. From an empirical standpoint, we observed that even prior to masking the policy exhibits proficiency in allocating the probability mass to legal actions in a given step, a phenomenon crucial for the successful performance of the hierarchical method.
>
> 3. It is true that in our study all agents adjust only one substation at a time. This is a constraint which is enforced by the power system environment (at least in all L2RPN competitions) in order to mimic real-world limitations (human operators usually cannot handle more than one substation within 5 minutes). This is also mentioned in Section 2.3.

---

> > ### Author Response · Authors · 2023-07-01
> > **Response to Review by Reviewer KGUv (part 2)**
> >
> > Other comments \& minor suggestions:
> > 1. It could indeed be good to elaborate this point more. In short: Yes, the low-level control (i.e. the choice of the substation configuration) is sensitive to long-term return. The reason is that even after a suitable substation has been chosen still a significant amount of different substation configurations is available. These can have very different effects on the overall topology and, hence, on how the load flow is routed. Consequently, also at the lowest level configurations can be sub-optimal in the short-term (i.e. when chosen greedily) and only optimal in the long-term.
> >
> > 2. We think it is a very good suggestion to compare our approach with the one in [2]. We could add the following text:
> >
> >     In [2] a hierarchical approach is presented which is complementary to the hierarchical approach presented in this study. More precisely, in [2] a higher-level RL controller chooses entire goal topologies at a single timestamp, that is, the configurations of all substations of the network are specified at a single timestamp. Choosing a goal topology can represent a temporally extended action due to the environment constraint that only one substation at a time can be reconfigured. Subsequently, in [2] a lower-level rule-based policy specifies the temporal sequence of the substation re-configurations which are pre-specified by the higher-level policy.
> >
> >     Consequently, differences between the two HRL approaches are that in [2] $(i)$ RL is applied only at one level (i.e. feedback-loops between RL policies are not investigated), $(ii)$ substation choice and configuration choice are done by the same policy (i.e. we dissect the action space more granularly), and $(iii)$ the latter is done partly in a temporally extended fashion (which implies that monitoring of changes in network state is partly neglected in these decisions). Moreover, we note that the specific role of HRL is less clearly analysed in [2] due to the simultaneous employment of several other advanced techniques like GNNs, transformers, and afterstates. It will be interesting to see future studies in which the two HRL approaches are compared and possibly combined, and in which the specific contribution of HRL in combination with other techniques is revealed.

---

> > ### Author Response · Authors · 2023-07-19
> > **Paper revised**
> >
> > We corrected the reward formula in Section 2.5.

---

### Comment · Action_Editors · 2023-07-25
**Please Read and  Respond to Author Response**

Hi,

If you haven’t already, please read and respond to the authors’ response to your initial review ASAP. The authors have made an effort to speak to your initial comments and concerns and it is important that their responses be taken into account.

Best,
AE

---

### Decision · Action_Editors · 2023-08-06

**Recommendation:** Reject

**Comment:**

The paper explores the use of hierarchical reinforcement learning (HRL) to mitigate the combinatorial complexity of controlling power networks. The proposed approach factorizes the action space across three levels: at the lowest level, actions modify the configuration of individual substations; intermediate actions determine which substation to modify at each time step; while actions at the highest level correspond to whether or not to make adjustments to the network. The paper evaluates the proposed HRL algorithm on simulated power network control problems of varying complexity and compares its performance to non-hierarchical RL and greedy baselines.

The formulation of power network control as a learning problem continues to attract attention within the machine learning community, which speaks to the relevance of the paper. While the formulation of power network control as a sequential decision making problem is not new, the paper's primary novelty lies in its three-level factorization of the action space via its framing of the problem in the context of hierarchical RL. The paper presents an evaluation of the advantages of this factorization with comparisons to both hand-crafted (greedy) strategies as well as "flat" RL in both a simple (single-scale) power network control setting as well as on a more realistic environment.

The paper received three reviews by researchers in reinforcement learning, who read the author responses as did the associate editor. The reviewers acknowledge the novelty with the way in which the paper factorizes the action space and the extent of the experimental evaluation. However, the reviewers question the extent of the paper's contributions with regards to the application of HRL to the problem of power network control. Specifically, the reviewers find that the paper lacks substantive insights into the utility of hierarchical approaches to power network control. The value of such a paper is not necessarily in showing that existing learning-based methods can be applied to an existing problem, but rather that doing so requires applying these methods in novel ways, or its value is in thorough experimental analyses that provide new insights into the problem. While the paper may be the first to apply hierarchical RL to power network control, the manner by which HRL is employed is straightforward. Meanwhile, the experimental results do not provide a compelling demonstration of the advantages of HRL over a "flat" PPO-based policy, which yields the best performance in the simpler domain (without contingencies) and second-best in the more challenging domain (with contingencies), and they lack an adequate analysis of the application of HRL to power network control. The associate editor appreciates the effort that the authors put into addressing the reviewers' concerns. The responses help to clarify the contributions as they relate to the application of HRL to power network control. The authors are encouraged to incorporate this discussion into any subsequent version of the paper.

**Audience:**

The formulation of power network control as a learning problem continues to attract attention within the machine learning community, which speaks to the relevance of the paper to a not insignificant number of ML researchers.

**Claims And Evidence:**

The reviewers acknowledge the novelty with the way in which the paper factorizes the action space and the extent of the experimental evaluation. However, the reviewers question the extent of the paper's contributions with regards to the application of HRL to the problem of power network control. Specifically, the reviewers find that the paper lacks substantive insights into the utility of hierarchical approaches to power network control. The value of such a paper is not necessarily in showing that existing learning-based methods can be applied to an existing problem, but rather that doing so requires applying these methods in novel ways, or its value is in thorough experimental analyses that provide new insights into the problem. While the paper may be the first to apply hierarchical RL to power network control, the manner by which HRL is employed is straightforward. Meanwhile, the experimental results do not provide a compelling demonstration of the advantages of HRL over a "flat" PPO-based policy, which yields the best performance in the simpler domain (without contingencies) and second-best in the more challenging domain (with contingencies), and they lack an adequate analysis of the application of HRL to power network control. The associate editor appreciates the effort that the authors put into addressing the reviewers' concerns. The responses help to clarify the contributions as they relate to the application of HRL to power network control. The authors are encouraged to incorporate this discussion into any subsequent version of the paper.